# Generation of pluripotent stem cell-derived mouse kidneys in *Sall1*-targeted anephric rats

Teppei Goto [1], Hiromasa Hara[1,6], Makoto Sanbo[1], Hideki Masaki[2], Hideyuki Sato[2], Tomoyuki Yamaguchi [2], Shinichi Hochi[3], Toshihiro Kobayashi[1,4], Hiromitsu Nakauchi[2,5] & Masumi Hirabayashi[1,4]

Regeneration of human kidneys in animal models would help combat the severe shortage of donors in transplantation therapy. Previously, we demonstrated by interspecific blastocyst complementation between mouse and rats, generation of pluripotent stem cell (PSC)-derived functional pancreas, in apancreatic *Pdx1* mutant mice. We, however, were unable to obtain rat PSC-derived kidneys in anephric *Sall1* mutant mice, likely due to the poor contribution of rat PSCs to the mouse metanephric mesenchyme, a nephron progenitor. Here, conversely, we show that mouse PSCs can efficiently differentiate into the metanephric mesenchyme in rat, allowing the generation of mouse PSC-derived kidney in anephric *Sall1* mutant rat. Glomerular epithelium and renal tubules in the kidneys are entirely composed of mouse PSC-derived cells expressing key functional markers. Importantly, the ureter-bladder junction is normally formed. These data provide proof-of-principle for interspecific blastocyst complementation as a viable approach for kidney generation.

---

[1] Center for Genetic Analysis of Behavior, National Institute for Physiological Sciences, Okazaki 444-8787 Aichi, Japan. [2] Division of Stem Cell Therapy, Institute of Medical Science, The University of Tokyo, Minato-ku 108-8639 Tokyo, Japan. [3] Faculty of Textile Science and Technology, Shinshu University, Ueda 386-8567 Nagano, Japan. [4] The Graduate University of Advanced Studies, Okazaki 444-8787 Aichi, Japan. [5] Institute for Stem Cell Biology and Regenerative Medicine, Department of Genetics, Stanford University School of Medicine, Stanford, CA 94305, USA. [6] Present address: Center for Molecular Medicine, Jichi Medical University, Shimotsuke 329-0498 Tochigi, Japan. Correspondence and requests for materials should be addressed to M.H. (email: mhirarin@nips.ac.jp)

Organ transplantation is among the most effective treatments to improve quality-of-life (QOL) in patients with end-stage renal disease (ESRD). However, a chronic shortage of donor kidneys leaves many patients with ESRD no choice, but to undergo continued dialysis treatment, associated with poor QOL, high medical costs and risk of complications. Currently in the USA, an estimated 95,000 patients are waiting for a kidney transplant, resulting in an 80% kidney demand over all other organs[1]. Generation of transplantable kidneys from pluripotent stem cells (PSCs), such as embryonic stem cells (ESCs) or induced pluripotent stem cells (iPSCs), is an attractive solution to this problem. However, despite advances in ex vivo generation of renal compartments from PSCs[2–5], generating three-dimensional (3D), functional, and size-matched kidneys from PSCs remains a significant challenge[6].

Blastocyst complementation is an innovative and potentially promising approach[7]: when blastocysts harvested from mutant animals lacking specific organs, are injected with PSCs, the entire organ generates from the PSCs, in the relevant developmental niche of the resultant chimeras. For instance, use of apancreatic $Pdx1^{mut/mut}$ mouse host blastocysts allows the generation of PSC-derived pancreas by blastocyst complementation, in allogenic, as well as in an interspecific setting between mouse and rat[8,9]. Furthermore, transplantation of islets from mouse PSC-derived pancreas generated in $Pdx1^{mut/mut}$ rats successfully restored blood glucose levels in diabetic mice, demonstrating a proof-of-concept for the use of PSC-derived organs for therapy[9].

For kidney generation, the anephric $Sall1^{mut/mut}$ model, devoid of functional kidneys, can provide a suitable developmental niche for PSC-derived cells. Kidney formation requires reciprocally inductive interactions between the metanephric mesenchyme, a nephron progenitor, and the ureteric bud. The metanephric mesenchyme further differentiates into the glomerular epithelia

and renal tubules. $Sall1$, an essential gene expressed in the metanephric mesenchyme, is crucial for ureteric bud attraction toward the mesenchyme in mice[10,11]. Thus, $Sall1^{mut/mut}$ mice show an anephric phenotype due to failed signaling at E11.5[10]. Mouse PSCs injected into the $Sall1^{mut/mut}$ mouse blastocysts, form exclusively PSC-derived metanephric mesenchyme which interacts with the ureteric bud, resulting in the generation of mouse-PSC-derived kidney[12]. In an interspecific setting, however, we previously reported that rat PSCs fail to form kidneys in $Sall1^{mut/mut}$ mice[12], despite developing chimeric renal tissues in wildtype mice[9]. This finding impedes testing the therapeutic potential of kidneys created in a xenogenic environment, as well as in addressing fundamental questions in biology such as size regulation of solid organs.

Here, we show, in an interspecific setting between mouse and rat, that unlike rat PSCs, mouse PSCs efficiently contribute to Sall1 positive metanephric mesenchyme. Therefore, we are able to successfully generate mouse kidneys in the $Sall1^{mut/mut}$ rat model by interspecific blastocyst complementation.

## Results

**Contribution of PSCs to metanephric mesenchyme in chimeras.** We first attempted to understand the causes behind the failure of interspecific blastocyst complementation, for kidney generation, when rat PSCs were injected into $Sall1^{mut/mut}$ mouse blastocysts. Since, during allogenic blastocyst complementation, wildtype PSC-derived cells could replace mutant cells in the metanephric mesenchyme[12], we reasoned that a required level of PSC contribution to the metanephric mesenchyme, is essential for the successful generation of the PSC-derived kidney. The metanephric mesenchyme expressing Sall1 and Six2, another nephron progenitor marker in mice[13,14], initiates the ureteric bud interaction

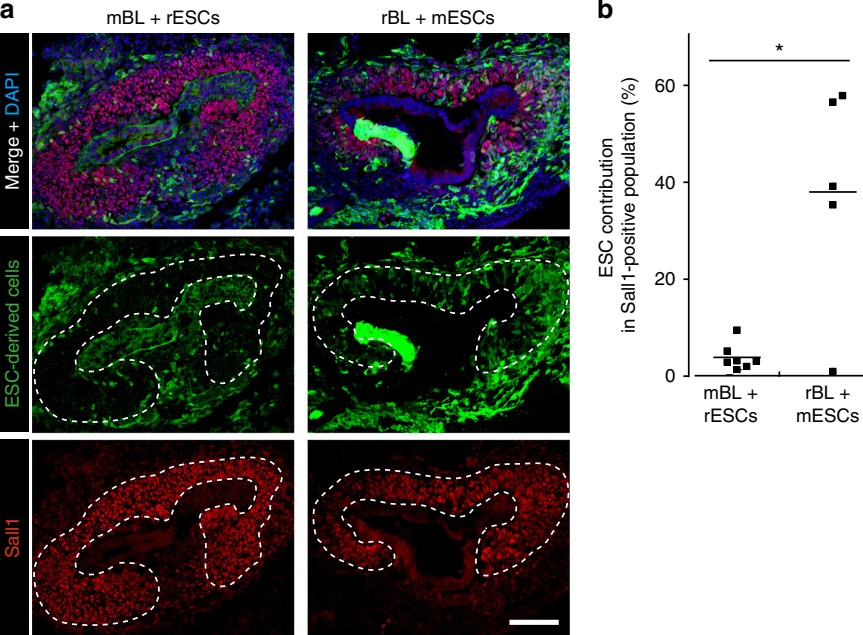

**Fig. 1** Fetal metanephric mesencyme in interspecific chimeras between mouse and rat. **a** Representative photomicrographs of contribution of ESCs (pseudocolored green) to Sall1-positive cells (red) at E11.5 in interspecific chimera by injection of rat ESCs into wildtype mouse blastocysts and at E13.5 in that of mouse ESCs into wildtype rat blastocysts. The dotted line indicates the metanephric mesenchyme. Chimeric mice were generated by microinjecting tdTomato-labeled rat ESCs, into mouse blastocysts. Chimeric rats were generated by microinjecting GFP-labeled mouse ESCs into rat blastocysts. mBL mouse blastocyst, rBL rat blastocyst, mESCs mouse embryonic stem cells, rESCs rat embryonic stem cells. Scale bar: 100 μm. **b** Quantification of the contribution of Sall1-positive cells in **a**. Horizontal bars represent the mean percentage. All data were obtained from 2 independent experiments. *$p < 0.05$ by two-tailed Student's $t$-test analysis. Source data are provided as a Source Data file

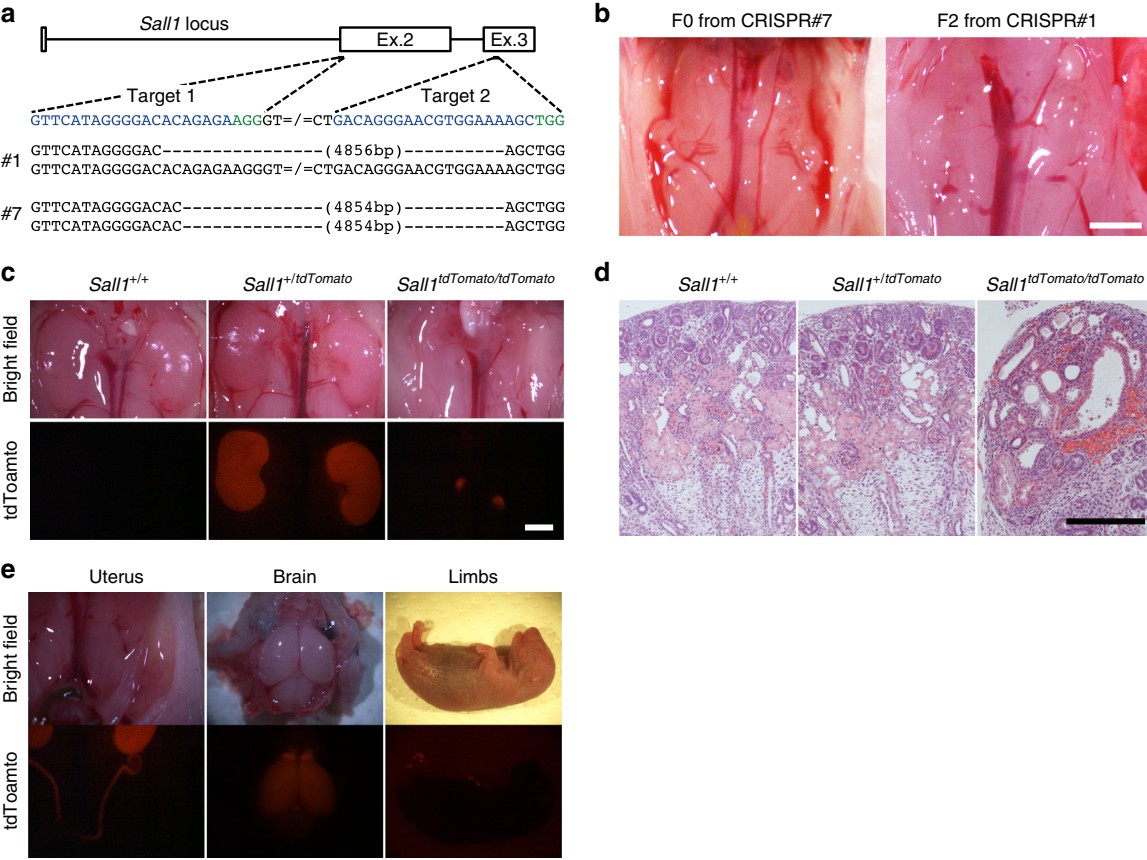

**Fig. 2** Generation of anephric *Sall1^mut/mut^* rats. **a** Design of CRISPR/Cas9 system to create a large deletion in the *Sall1* gene and the deletions in founder rats (CRISPR#1 and CRISPR#7). Blue and green letters indicate target and PAM sequences, respectively. **b** Anephric phenotype in CRISPR#7 founder and F2 offspring of the CRISPR#1 line. Long region deletion contributed to stable generation of anephric rats. Scale bar: 2 mm. **c** Kidneys in *Sall1^+/+^*, *Sall1^+/tdTomato^* and *Sall1^tdTomato/tdTomato^* rats. Scale bar: 2 mm. **d** Hematoxylin and eosin-stained kidney sections in *Sall1^+/+^*, *Sall1^+/tdTomato^*, and *Sall1^tdTomato/tdTomato^* rats. Scale bar: 200 μm. **e** Sall1-tdTomato expression in uterus, brain and limbs in *Sall1^+/tdTomato^* rat. ESC clone #4 was used to establish heterozygous mutant line HR#4

at E11.5 in mouse and E13.5 in rat (Supplementary Fig. 1). Thus, we investigated a contribution of donor PSCs to the metanephric mesenchyme after injection of rat ESCs into wildtype mouse blastocysts (R- > M) and mouse ESCs into wildtype rat blastocysts (M- > R) (Fig. 1a). Quantitative immunohistological analysis revealed that, in the Sall1 positive metanephric mesenchyme, the contribution of rat ESCs in R- > M was significantly lower than that of mouse ESCs in M- > R (3.9 vs. 38.0%) (Fig. 1b). Notably, in R- > M, the contribution of rat ESCs to Sall1-negative surrounding tissues including the ureteric bud was not as low as the metanephric mesenchyme (Fig. 1a). Indicating that, rat PSCs are less efficient in differentiating specifically into the metanephric mesenchyme compared to other tissues in the mouse. These data suggest that, as in the previous study[12], the number of rat PSC-derived metanephric mesenchyme in *Sall1^mut/mut^* mouse might not be sufficient to attract the ureteric bud for renal development.

**Generation and characterization of *Sall1^mut/mut^* rats.** Since mouse PSCs efficiently contribute to metanephric mesenchyme in M- > R in contrast to rat PSCs in R- > M, we believed we could generate mouse PSC-derived kidney in anephric rats by inter-specific blastocyst complementation. To provide the necessary developmental niche for a kidney in the rat, we first created a *Sall1^mut/mut^* rat model. We microinjected clustered regularly interspaced short palindromic repeats (CRISPR)/Cas9 constructs

into rat zygote pronuclei, to delete an approximately 4.9 kb region containing exon 2 and exon 3 of rat *Sall1* gene, and generated heterozygous founder rats (Fig. 2a, Supplementary Fig. 2a–c). After intercrossing heterozygous mutant rats, wild-type (*Sall1^+/+^*), heterozygotes (*Sall1^+/mut^*), and homozygous mutant (*Sall1^mut/mut^*) zygotes developed to embryonic day 21.5 (E21.5) fetuses in a Mendelian fashion. Notably, the *Sall1^mut/mut^* F0 fetus (CRISPR #7), as well as all of the *Sall1^mut/mut^* F2 fetuses from CRISPR #1 line intercross exhibited a kidney-deficient (anephric) phenotype (Fig. 2b), similar to *Sall1^mut/mut^* mice[10]. To visualize the expression of Sall1 in rats, we also generated *Sall1-tdTomato* knock-in rats by replacing the *Sall1* gene with *tdTomato* via homologous recombination (HR)-based gene targeting in rat ES cells (Supplementary Fig. 3a–d). While *Sall1^tdTomato/tdTomato^* neonates exhibited an anephric phenotype like *Sall1^mut/mut^* rats, *Sall1^+/tdTomato^* neonate kidneys uniformly expressed tdTomato (Fig. 2c). Hematoxylin and eosin staining revealed the presence of a rudimentary kidney and intact ureter without functional nephric units in *Sall1^tdTomato/tdTomato^* rats, whereas normal kidney histology was observed in *Sall1^+/tdTomato^* and *Sall1^+/+^* rats (Fig. 2d). In addition to the renal parenchyma, the expression of tdTomato was also observed in the brain, uterus, and limbs in both *Sall1^tdTomato/tdTomato^* and *Sall1^+/tdTomato^* neonates (Fig. 2e), which corresponds to the Sall1 expression pattern in mice[10,15]. Collectively, these data indicate *Sall1* gene is well conserved between mouse and rat in its essential role in

**Table 1 Result of interspecific blastocyst complementation for kidney generation**

|  | Blastocyst injected | Living fetuses | Chimeric fetuses | Genotype | | | Kidney complemented |
|---|---|---|---|---|---|---|---|
|  |  |  |  | Wildtype | Hetero | Homo |  |
| HR#4 | 99 | 60 (61%) | 39 (65%) | 19 | 15 | 5 | 2 (40%) |
| CRISPR#1 | 153 | 50 (33%) | 37 (74%) | 8 | 16 | 13 | 10 (77%) |

kidney development, as well as in its tissue-wide expression pattern. Importantly, the anephric $Sall1^{mut/mut}$ (including $Sall1^{tdTomato/tdTomato}$) rat model can now provide the necessary developmental niche to generate a kidney.

**Generation of mouse kidney in rat**. We then performed interspecific blastocyst complementation using $Sall1^{mut/mut}$ rat blastocysts with green fluorescent protein (GFP)-labeled mouse ESCs to generate mouse kidney in the rat. We injected 252 blastocysts derived from the intercross of $Sall1^{+/mut \, or \, tdTomato}$ (CRISPR#1, HR#4) with mouse ESCs and transferred them into pseudopregnant rat uteri. A total of 76 of 110 fetuses were identified as interspecific chimeras, assessed by GFP fluorescence in the perinatal skin at E20.5–21.5 and P0. Retrospective genotype analysis revealed that $Sall1^{+/+}$, $Sall1^{+/mut}$, and $Sall1^{mut/mut}$ chimeric rats had been born with a Mendelian ratio (Table 1). The mouse ESCs contributed to various organs including skin, muscle, heart, thymus, and kidney, but contributed less to the pancreas, liver, and lung regardless of genotyping (Fig. 3a). Average chimerism in splenic lymphocytes was 3.0%, independent of genotype (Fig. 3b). In the kidney, all of the $Sall1^{+/mut}$ and $Sall1^{+/+}$ chimeric neonates had chimeric kidneys composed of both donor- and host-derived cells, remarkably, of the 18 $Sall1^{mut/mut}$ chimeric neonates, 12 (67%) had a pair of kidneys uniformly expressing GFP, indicating successful generation of mouse-PSC derived kidneys in the rats (Fig. 3c). The rest of the $Sall1^{mut/mut}$ chimeric neonates without kidneys showed low levels of chimerism in lymphocytes (red squares in Fig. 3b). Thus, as in the $Sall1^{mut/mut}$ mouse complemented with rat PSCs, low levels of chimerism, likely correlate with failure to form PSC-derived kidneys. Furthermore, we successfully generated kidneys derived from allogenic rat ESCs (Supplementary Fig. 4a–c), as well as xenogenic mouse iPSCs (Supplementary Fig. 4d–f), demonstrating the consistency of blastocyst complementation in kidney, regardless of the donor cell types. While mouse ESC-derived kidneys in $Sall1^{mut/mut}$ rats at P0 were histologically normal (Fig. 3d), their size was smaller than normal neonatal and $Sall1^{+/mut}$ and $Sall1^{+/+}$ chimeric rats, and were similar to that of mice kidneys (Fig. 3e). Thus, we hypothesize that at least until birth, the size of the kidney might either be under the control of certain species-specific intrinsic factor(s), or reflect the extent of chimerism of donor PSCs.

We next investigated whether the $Sall1^{mut/mut}$ chimeric fetuses survived to adulthood. We injected 148 blastocysts derived from intercrossing $Sall1^{+/mut}$ rats with mouse ESCs; 63 of 100 offspring were GFP-positive interspecific chimeras. Only 21 of the offspring survived for more than 8 weeks. Notably, all of the 21 rats that survived to adulthood were either $Sall1^{+/+}$ (n = 9) or $Sall1^{+/mut}$ (n = 12), whereas none of the $Sall1^{mut/mut}$ rats with mouse PSC-derived kidneys survived much longer after birth due to defects in milk suckling (Supplementary Fig. 5a), this phenotype corresponds to $Sall1^{mut/mut}$ mice with mouse-PSC-derived kidneys[10].

**Characterization of mouse PSC-derived kidney in rat**. We performed immunohistology to validate correct expression of

functional markers of renal components and to identify the origin of the cells in the mouse-ESC derived kidneys in $Sall1^{mut/mut}$ rats, at P0. We distinguish each renal component based on the specific functional markers (Supplementary Fig. 6a, Supplementary Table 1). In $Sall1^{+/+}$ and $Sall1^{+/mut}$ chimeric rat kidneys, all the renal components consisted of a mixture of mouse and rat-derived cells (Fig. 4a, Supplementary Fig. 6b). In contrast, mouse PSC-derived kidneys in $Sall1^{mut/mut}$ chimeric rats, were entirely composed of GFP-positive mouse ESCs-derived cells, including the Nephrin[16]-positive podocytes, the Aquaporin 1[17]-positive proximal tubule, descending limb of loop of Henle, and loop of Henle, and the Na$^+$/K$^+$ ATPase α-1[18] strongly positive ascending limb of loop of Henle and distal tubule (Fig. 4a). Correct localizations of these markers (Nephrin at glomerular basement membrane; Aquaporin 1 at both apical and basolateral plasma membrane; Na$^+$/K$^+$ ATPase α-1 at basolateral plasma membrane) were detected. Notably, these markers are strongly associated with the functioning of the nephron, suggesting mouse PSCs differentiate correctly into functional cells in the xenogenic environment. Additionally, from the histological analysis, we found the Bowman's capsular epithelial cells formed outside the podocytes were also entirely mouse ESC derived (arrowheads in Fig. 4a **upper right**). On the other hand, the Calbindin-positive collecting ducts in the renal medulla, and CD31-positive capillary vessels of the glomeruli and cortical blood vessels consisted of both mouse and rat derived cells (Fig. 4b). These results indicate that all the metanephric mesenchyme-derived cells impaired in $Sall1^{mut/mut}$ rats originate from mouse PSCs, whereas the other compartments in the kidney comprise both mouse PSCs and $Sall1^{mut/mut}$ rat blastocysts (Fig. 4c). The number of glomeruli in kidney was similar between $Sall1^{mut/mut}$ chimeric rats and control mice (Supplementary Fig. 7a, c), while the size of glomeruli was similar between $Sall1^{mut/mut}$ chimeric rats and control rats (Supplementary Fig. 7b).

We next investigated whether mouse ESC-derived kidneys can form proper ureter and bladder connections. An intraurethral dye infusion proved patency between the ureter and the bladder in two of the four P0 $Sall1^{mut/mut}$ chimeras harboring mouse PSC-derived kidneys (Fig. 4d), indicating the organ has the potential to excrete urine.

Finally, to check if the filtering system in the kidneys work, we measured blood urea nitrogen (BUN) and creatinine (CRE) in the serum of $Sall1^{mut/mut}$ chimeric rats with mouse PSC-derived kidney, 10–14 h after birth. No significant difference compared with $Sall1^{+/+}$ and $Sall1^{+/mut}$ chimeras, as well as $Sall1^{mut/mut}$ rats lacking kidneys was observed (Supplementary Fig. 5b), possibly due to inadequate activity of the neonates postnatally.

**Discussion**

Our work demonstrates the successful generation of mouse PSC-derived kidneys in $Sall1^{mut/mut}$ rats, by interspecific blastocyst complementation. This approach has succeeded for pancreas[8,9] and thymus[19] generation between mouse and rats. However, applying interspecific blastocyst complementation for kidney

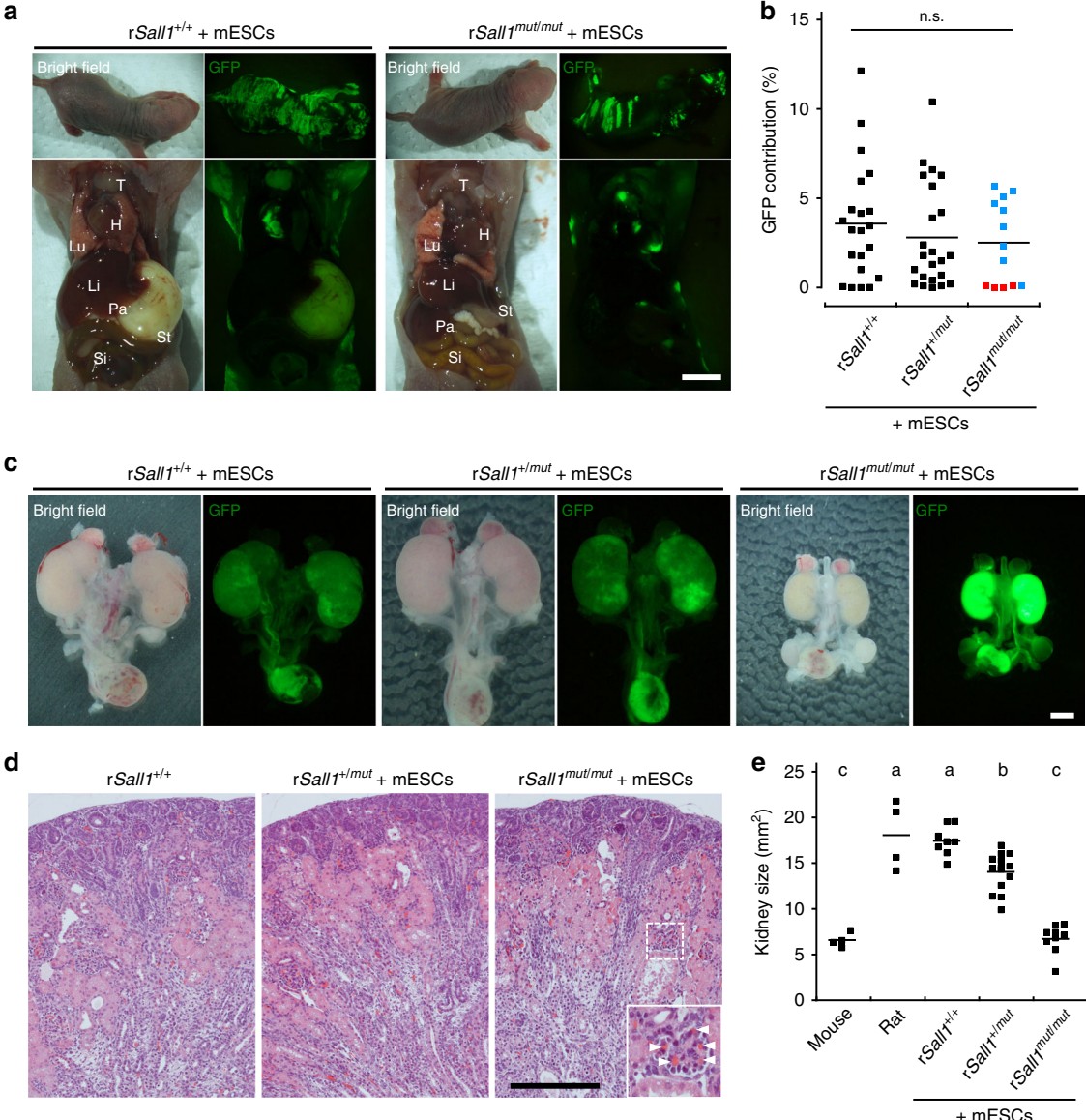

**Fig. 3** Generation of mouse ESCs-derived kidney in *Sall1mut/mut* rats by interspecific blastocyst complementation. **a** Contribution of GFP-labeled mouse ESCs to various organs in interspecific chimera at after birth. T: thymus; H: heart; Lu: lung; Li: liver; Pa: pancreas; SI: small intestine; St: stomach. Scale bar: 2 mm. **b** Variability of chimerism in splenic lymphocytes. Horizontal bars represent the mean percentage. Blue and red squares for *Sall1mut/mut* represent nephric and anephric phenotype, respectively. All data were obtained from 5 independent experiments. n.s.: Not statistically significant by one-way ANOVA. **c** GFP fluorescence derived from mouse ESCs in kidneys of *Sall1+/+*, *Sall1+/mut*, and *Sall1mut/mut* chimeras, respectively. Scale bar: 2 mm. **d** Normal histology of ESC-complemented kidneys in *Sall1mut/mut* chimera, compared with *Sall1+/+*, *Sall1+/mut* chimeras, confirmed by hematoxylin and eosin staining. Inset shows glomeruli, and arrowheads indicate red blood cells. Scale bar: 200 μm. **e** Quantification of kidney size of *Sall1+/+*, *Sall1+/mut*, and *Sall1mut/mut* chimeras, respectively. Horizontal bars represent the mean percentage. All data were obtained from 3 independent experiments. Different letters (a, b, and c) indicate statistically significant differences ($p < 0.05$) based on a one-way ANOVA followed by Tukey's HSD post hoc test. Source data are provided as a Source Data file

genesis would have a more substantial impact on regenerative medicine, due to high donor demand in end-stage renal disease patients[20,21].

A key finding that we report is that sufficient contribution of xenogenic PSCs to the metanephric mesenchyme is essential for kidney formation. Histological analysis at the onset of metanephric mesenchyme and ureteric bud interaction revealed that contribution of rat PSCs to Sall1 positive metanephric mesenchyme in R- > M is less efficient than mouse PSCs in M- > R, which likely explains the failure of kidney formation in *Sall1mut/mut* mice complemented with rat PSCs[12]. Also in M- > R, low chimerism of mouse PSCs is highly correlated with failure of complementation in *Sall1mut/mut* rats. Thus, in the kidney, a minimum level of PSC contribution to the metanephric mesenchyme is required to initiate interaction with the ureteric bud. Since PSCs contribute to all other tissues to a similar extent in an allogenic setting, the interspecific variability for kidney formation is likely due to incompatibilities in environmental cues such as cytokines and extracellular matrix during development, rather than potency of the PSCs. As we have shown previously, in interspecific chimeras between mouse and rat, the level of contribution of the xenogenic PSCs varies from organ-to-organ[8,22]. What factors specifically promote efficient PSC contribution

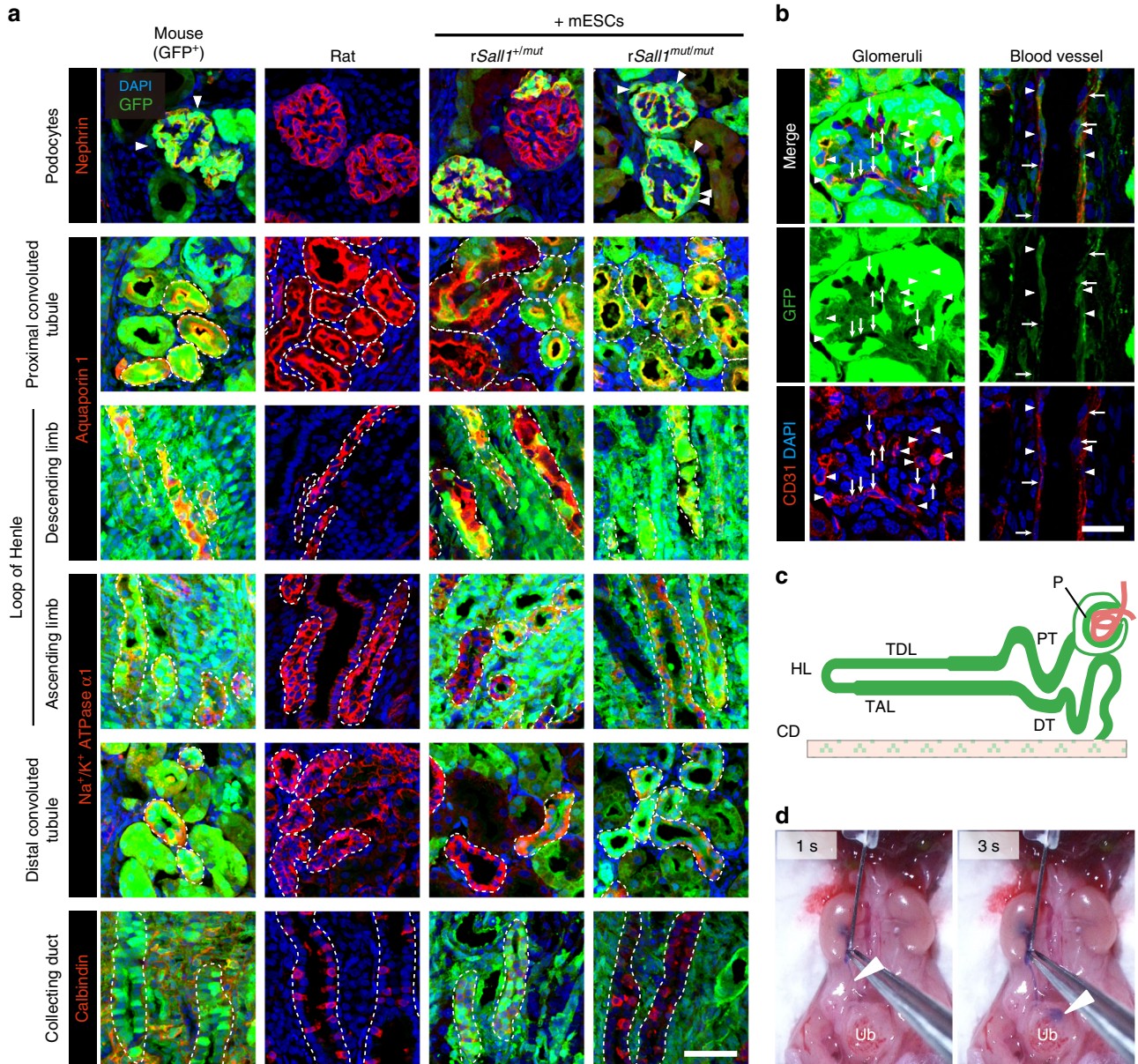

**Fig. 4** Characterization of mouse ESC-derived kidney generated in *Sall1^{mut/mut}* rats. **a** Immunohistochemical staining of neonatal kidneys for GFP (green) with markers for each renal component (red); Nephrin for Podocytes, Aquaporin 1 for proximal convoluted tubule in the renal cortex and thin descending limb/Henle's loop in the renal medulla, Na$^+$/K$^+$ ATPase α-1 for thin ascending limb in the renal medulla and distal convoluted tubule. Calbindin for collecting tubule in the renal medulla. The nuclei were stained with DAPI (blue). Nephrin localizes a linear pattern along the glomerular basement membrane. Aquaporin 1 exists in both apical and basolateral plasma membranes of proximal tubules and descending thin limbs. Na$^+$/K$^+$ ATPase α-1 exists in basolateral plasma membranes of thin ascending limbs and distal convoluted tubules. Scale bar: 50 μm. **b** Immunohistochemical staining of mouse-ESC derived neonatal kidneys in *Sall1^{mut/mut}* rats for GFP (green) with vascular endothelium, CD31 (red). The nuclei were stained with DAPI (blue). Scale bar: 25 μm. **c** Schematics of nephron and collecting tubule in complemented kidney in *Sall1^{mut/mut}* rats. Green shows the contribution of mouse ESCs. P: Podocytes; PT: Proximal convoluted tubule; TDL: Thin descending limb of Henle's loop; HL: Henle's loop; TAL: Thick ascending limb of Henle's loop; DT: Distal convoluted tubule; CD: Collecting duct. **d** Ureter–bladder junction formation confirmed by intraurethral dye infusion in E22.5 *Sall1^{mut/mut}* rats with mouse PSC-derived kidneys. Arrowheads represent the lead position of trypan blue solution 1 and 3 s after injection. Ub: urinary bladder

remains unknown at this time. Nevertheless, the extent of contribution of xenogenic PSCs to the target organ, specifically the tissues or lineages to be replaced, should be an important consideration while choosing the appropriate host during interspecific blastocyst complementation.

The kidney generated by blastocyst complementation in our study is likely to function in vivo, based on expression of functional markers in the mouse PSC-derived nephron and the ureter-bladder junction. Additionally, we observed patency between ureter and bladder, which indicated normal development of the kidney, with potential to excrete urine. Although testing the filtering system in nephrons would strengthen evidence for its function, postnatal lethality of *Sall1^{mut/mut}* rat containing mouse PSC-derived kidney limits further investigation. Despite the presence of wildtype PSC-derived cells in chimeras, the milk suckling defect observed in *Sall1^{mut/mut}* rats was not rescued, akin

to *Sall1^(mut/mut)* mouse with mouse PSC-derived kidney[12]. Sall1 is highly expressed in the olfactory bulb. Sall1 mutant olfactory bulbs are small in size, show abnormalities in neurogenesis and have reduced mitral cells, likely rendering the newborn mice anosmic and therefore unable to suckle milk[23,24]. To overcome the limitations posed by the mutant host phenotype, disrupting other essential genes for kidney development should be examined. Alternately, conditionally deleting essential genes or ablating specific cells might be an avenue to explore.

For transplantation therapy, further elimination of host blastocysts derived cells would be essential to reduce the use of immunosuppressive treatment. While all the metanephric mesenchyme cells were of mouse PSC origin in *Sall1^(mut/mut)* rats, the ureteric bud-derived collecting tubes and blood vessels were composed of a mixture of donor and host cell types. Thus, a combination of genetically modified animals, which can provide the appropriate developmental niche, should be considered for complete replacement of the kidney by PSC-derived cells.

In the future, it might be possible to generate human PSC-derived organs in livestock by using the blastocyst complementation approach. Despite successful contribution of human PSCs to pig embryos after blastocyst injection[25], the chimerism was low and insufficient for organ development, like our observations for the kidney. Exploring alternative culture conditions of donor human PSCs or inhibition of apoptosis by forced expression of anti-apoptotic factors to prevent the elimination of PSC-derived cells might be helpful to increase chimerism[26–28]. Conversely, increasing interspecific chimerism associated with poor survival of the chimeras[8,25], and raises ethical concerns regarding the presence of human cells in neural or germline lineages[29]. Thus, limiting the potency towards a specific lineage by genetic manipulation in PSCs[30], or context-dependent usage of differentiated cells from PSCs upon inhibiting apoptosis[26,31], may prove to be viable options to consider.

In conclusion, we have extended the application of interspecific blastocyst complementation between mouse and rat to generate fully developed kidneys. This study provides a tool to address fundamental questions on kidney ontogeny and regulation of size in solid organs and could offer clues to successful generation of human organs in animals.

## Methods

**Experimental statement.** All procedures for animal experimentation were reviewed and approved by the Animal Care and Use Committee of the National Institute for Physiological Sciences. Specific pathogen-free Crlj:WI rats (RGD ID: 2312504) were purchased from Charles River Laboratories Japan, Inc. (Kanagawa, Japan). Specific pathogen-free C57BL/6 NCrSlc and C57BL/6N-Tg (CAG-EGFP)[32] mice were purchased from Japan SLC, Inc. (Shizuoka, Japan). All rats and mice were housed in an environmentally controlled room with a 12-h dark/12-h light cycle at 23 ± 2 °C and humidity of 55 ± 5%, and given ad libitum access to a laboratory diet (CE-2; CLEA Japan, Inc., Tokyo, Japan) and filtered water. Study reagents were purchased from Sigma-Aldrich Corp. (St. Louis, MO, USA) unless otherwise stated.

**Generation of tdTomato-knockin rats at *Sall1* locus.** The targeting vector was constructed from a bacterial artificial chromosome (BAC) clone RNB1-261M04[33] using a counter-selection BAC modification kit and a BAC subcloning kit (Gene Bridges Gmbh, Heidelberg, Germany). The RNB1-261M04 was provided by RIKEN BRC through the National Bio-Resource Project of MEXT, Japan. In brief, *tdTomato* (Clontech, Mountain View, CA, USA) with FRT-PGK-gb2-neo-FRT cassette (Gene Bridges) was substituted at the site between the second and third exons located within functional DNA binding domains of the *Sall1* gene (accession no. NM_001107415.2 (https://www.ncbi.nlm.nih.gov/nuccore/NM_001107415.2)). The BAC backbone was replaced with a plasmid cassette containing a chloramphenicol resistance gene and *ori* from pACYC184 DNA (Nippon Gene Co., Ltd., Toyama, Japan). The final targeting vector was linearized by ScaI digestion.

Rat ESCs (WDB/Nips-ES1/Nips, RGD ID: 10054010) were cultured on mitomycin C (MMC)-treated Neo-MEFs (KBL9284100; Oriental Yeast Co., Ltd.

Tokyo, Japan) in 2iF medium consisting of N2B27 medium with 1-μM mitogen-activated kinase kinase (MEK) inhibitor PD0325901 (Stemgent Inc., Cambridge, MA, USA), 3-μM glycogen synthase kinase 3 (GSK3) inhibitor CHIR99021 (Axon Medchem, Groningen, The Netherlands), 1000 U/mL ESGRO® (Merck Millipore, Burlington, MA, USA), and 10-μM forskolin. The targeting vector (25 μg) was introduced into 5 × 10^6 ESCs by electroporation at 800 V, 10 μF in 500 μL of N2B27 medium, and the electroporated ESCs were treated with 200-μg/mL G418 for 48 h. Knockin at the *Sall1* locus was determined by PCR for *tdTomato* with AmpliTaq® DNA polymerase (Applied Biosystems Inc., Foster City, CA, USA) and for the 5′-upstream HR region of *Sall1* locus with PrimeSTAR GXL DNA polymerase (Takara Bio Inc., Shiga, Japan), then confirmed by southern blot using DIG-labeled probes. E4.5 blastocysts were retrieved from Crlj:WI female rats, injected with recombinant ESCs using a piezo-driven micromanipulator (Prime Tech Ltd., Ibaraki, Japan), and transferred into uteri of pseudopregnant Crlj:WI females at 3.5 days post-coitum (dpc) to generate chimeric rats. F1 heterozygous offspring were derived from wildtype Crlj:WI females mated with male chimeras (. Genomic DNA samples were extracted from ear tissues, then used for PCR screening and genotyping in *Sall1* locus. Primer sequences used for PCR screening, Southern blotting, and genotyping are listed in Supplementary Table 2.

**Generation of *Sall1*-knockout rats by CRISPR/Cas9 system.** Pronuclear-stage zygotes were harvested from 7–8-week-old Crlj:WI females that had been super-ovulated by intraperitoneal injections of 150-IU kg⁻¹ equine chorionic gonado-tropin (eCG; Aska Pharmaceutical Co., Ltd., Tokyo, Japan) and human chorionic gonadotropin (hCG; Aska Pharmaceutical) at an interval of 48–50 h and had mated with fertile Crlj:WI males. Two guide sequences targeted at the second and third exons of rat *Sall1* locus (target 1 and target 2: Fig. 2a) were introduced into bi-cistronic expression vector px330 expressing Cas9 and single-guide RNA. The px330 vectors with one of the two guide sequences (2.5 ng μL⁻¹ each) were co-injected into the pronucleus of zygotes. The zygotes were cultured for 16 h in mKRB medium at 37 °C under 5% CO₂ in air, then evaluated for survival and cleavage to the 2-cell stage. All the surviving embryos were transferred into ovi-ductal ampullae of pseudopregnant females at 0.5 dpc. Living fetuses were recovered from the recipient females at E21.5, or full-term offspring were delivered by the recipients. Genomic DNA was extracted from ear tissues of founder rats (#1–#12), and used for PCR screening and genotyping in *Sall1* locus (Supplementary Table 2).

**Generation of chimeras with PSCs.** Mouse ESCs and iPSCs with GFP fluorescent marker (SGE2[12] and GT3.2[9]), or rat ESCs with Venus fluorescent marker (Crlj:WI-ES1/Nips, RGD ID: 10053737) and tdTomato fluorescent marker (WDB-Rosa26^(em1(RT2)Nips)-ES2/Nips, RGD ID: 10054032)[34], were used for blastocyst injection. For interspecific chimera generation, host blastocysts were obtained from Crlj:WI female rats (4.5 dpc) and C57BL/6NCrSlc female mice (3.5 dpc), respectively. For blastocyst complementation, heterozygous *Sall1*-KO females among HR#4 or CRISPR#1 rat lines were mated with heterozygous *Sall1*-KO males of the corresponding line, and the *Sall1^(+/mut)* × *Sall1^(+/mut)* rat blastocysts were retrieved at 4.5 dpc. Each of the seven PSCs that had been maintained on MMC-treated Neo-MEFs in the 2iF medium and isolated by trypsinization were microinjected into the blastocoel cavity of the blastocysts. Following a short-term recovery culture for 1–2 h, the injected blastocysts were transferred into uteri of pseudopregnant Crlj:WI females at 3.5 dpc (10–16 blastocysts per recipient). Fluorescence-based screening of chimeric rats was performed using E20.5–21.5 fetuses and P0 neonates, as well as genotype, chimerism and nephrogenetic phenotype analyses. Otherwise, the surrogate mothers were allowed to deliver, and the newborn offspring rats were analyzed for fluorescent marker expression, genotype, chimerism and phenotype either immediately or 8 weeks after birth.

**Analysis of genotype and chimerism.** Genomic DNA of chimeric rats was extracted from liver pieces or fluorescent-negative fraction of splenic lymphocytes sorted by fluorescence-activated cell sorting (FACS, SH800; Sony Corp., Tokyo, Japan). For HR#4 rat line, the *Sall1* locus was amplified by PCR using F1, R1 and R2 primers (Supplementary Table 2). For CRISPR#1 rat line, the *Sall1* locus was amplified by PCR using F1, R2 and R3 primers (Supplementary Table 2). Splenic lymphocytes were used to determine the fluorescence-based chimerism of each xenogeneic or allogeneic chimeric rat. The fluorescence-positive gates were set using the FACS histograms of the non-chimera samples (negative controls).

**Histological analysis.** Embryos and kidneys were fixed with 10% formalin neutral buffer solution or 4% paraformaldehyde and embedded in paraffin or in optimal cutting temperature (OCT) compound. Paraffin sections were deparaffinized with xylene, hydrated with graded ethanol, then HE-stained for light microscopy. Frozen sections with OCT compound were incubated with blocking buffer for 1 h at room temperature, and further incubated with primary antibody overnight at 4 °C and secondary antibody for 1 h at room temperature. Primary antibodies used were anit-Sall1 polyclonal antibody (rabbit IgG, 1:100; ab31526, Abcam plc., Cambridge, UK)[35], anit-Six2 polyclonal antibody (rabbit IgG, 1:100; 11562-1-AP, Proteintech Group, Inc., Rosemont, IL 60018, USA)[35], anti-GFP polyclonal

antibody (Chick IgY, 1:1000 dilution; ab13970, Abcam), anti-DsRed polyclonal antibody (rabbit IgG, 1:200 dilution; 632496, Takara Bio Inc.), anti-Nephrin polyclonal antibody (goat IgG, 1:200 dilution; AF3159-SP, R&D systems Inc., Minneapolis, MN, USA), anti-Podocin polyclonal antibody (rabbit IgG, 1:25 dilution; P0372, Sigma-Aldrich), anti-Calbindin polyclonal antibody (mouse monoclonal IgG, 1:200 dilution; ab82812, Abcam), anti-Aquaporin 1 polyclonal antibody (rabbit IgG, 1:50 dilution; AB2219, Merck Millipore), anti-Na$^+$/K$^+$ ATPase α-1 monoclonal antibody (mouse IgG, 1:50 dilution; 05-369, Merck Millipore) and anti-CD31 polyclonal antibody (rabbit IgG, 1:50 dilution; ab28364, Abcam). The secondary antibodies used for visualization were conjugated with Alexa488, Alexa546 or Alexa647 (1:300 dilution; Thermo Fisher Scientific Inc., Waltham, MA USA). After treatment with the appropriate antibodies, the sections were stained with 4',6-diamidino-2-phenylindole (DAPI) (Sigma-Aldrich) for nuclear visualization. The contribution of xenogenic PSCs to Sall1 positive metanephric mesenchyme was quantified by Fiji software.

**Biochemical assays in serum**. The serum levels of blood urea nitrogen (BUN) and creatinine (CRE) were measured by the Nagahama Institute for Biochemical Science at Oriental Yeast Co., Ltd. (Shiga, Japan). BUN was examined by the urease-glutamate dehydrogenase method, and CRE by the creatininase method, respectively.

**Statistical analysis**. The contribution of xenogenic PSCs to Sall1 positive metanephric mesenchyme was analyzed by two-tailed Student's $t$-test. The chimerism of GFP- or venus-positive rats, kidney size, and the serum levels of BUN and CRE were analyzed by one-way ANOVA followed by Tukey's HSD post hoc test with js-STAR (http://www.kisnet.or.jp/nappa/software/star/), and differences among genotypes were considered to be significant at $p < 0.05$.

**Reporting summary**. Further information on experimental design is available in the Nature Research Reporting Summary linked to this article.

## Data availability

The data that support the findings of this study are available from the corresponding author upon reasonable request. The source data underlying Figs 1b and 3c, f and Supplementary Figs 4c, f, 5b and 7a, b are provided as a Source Data file. A reporting summary for this Article is available as a Supplementary Information file.

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

## Acknowledgements

We thank members of the Hirabayashi lab, Keiko Yamauchi, Megumi Hashimoto, and Naoko Niizeki for help with animals, Reiko Terada and Fumika Yoshida for help with cell culture and molecular biology, Minako Onishi for secretarial support. We also thank Dr. Roopsha Sengupta for editing and providing critical input to the manuscript. We thank Spectrography and Bioimaging Facility, NIBB Core Research Facilities for technical support. This work was supported by grants from LEAP-AMED (JP18gm0010002) and Japan Science and Technology Agency, Exploratory Research for Advanced Technology to H.N. and M.H., Grant-in-Aid for Scientific Research from the Japan Society for the Promotion of Science (18H02367) to M.H. and T.K., the California Institute for Regenerative Medicine (LA1-06917) to H.N., and the NINS program for cross-disciplinary science study to T.G.

## Author contributions

M.H. and H.N. supervised the project. H.N. developed the concept. T.G. and M.H. designed the study. T.G., H.H., M.S., and M.H. generated biological data sets. T.G., H.M., H.S., and T.Y. generated histological data sets. T.G., S.H., T.K., and M.H. analyzed the data and wrote the manuscript.

## Additional information

**Competing interests:** The authors declare no competing interests.

