## [Peer Review File · Nature Communications]

Reviewers' Comments:

Reviewer #1:

Remarks to the Author:

This manuscript reports the generation of *Sall1* mutant rats with impaired nephrogenesis and the use of blastocyst complementation with GFP labeled mouse embryonic stem (ES) cells to generate fetal 3D kidneys with an intact ureter-bladder connection. Nephrogenesis was detected by the presence of podocyte and tubulus-specific markers. The analysis of E21.5 *Sall1*^{-/-} chimeric fetuses showed that the glomerular podocytes and proximal convoluted tubules originated from mouse ES cells, while the other parts of the kidney still contained rat cells. This limits the use of this approach for generating kidney grafts escaping immune rejection.

Although kidney formation was detected after xenogeneic blastocyst complementation in *SALL1* deficient fetuses, postnatal development of the chimeric animals was apparently hampered, since none of the surviving chimeras had the *Sall1*^{-/-} genotype. The authors speculate that other effects (not related to kidney development) of *Sall1* deficiency, such as impaired suckling may be responsible for postnatal lethality of blastocyst complemented *Sall1*^{-/-} chimeras. A more thorough analysis of postnatal *Sall1*^{-/-} chimeras would be required rather than just mentioning remnants of one chimeric offspring retrieved at postnatal day 4.

A previous study (Usui et al., *Am J Pathol* 180, 2417–2426, 2012) reported that it was not possible to generate in *SALL1* deficient mice kidneys from pluripotent rat stem cells. The authors speculated that key molecules involved in the interactions of the metanephric mesenchyme and ureteric buds do not cross-react between mouse and rat. The present study raises the question why it works – at least to some extent – the other way round. Potential mechanisms should be discussed.

Since the introduction of the manuscript starts with the problem of organ transplantation, the discussion should include potential pathways how the approach might be further developed towards clinical application. This would require host embryos from larger species, such as pig. Previous studies involving complementation of porcine blastocysts with human pluripotent stem cells showed very low chimerism. Are there strategies how to overcome this problem?

Reviewer #2:

Remarks to the Author:

The authors describe that mouse embryonic stem cells injected into rat blastocysts can rescue kidney development when the blastocysts are derived from *Sall1*-knockout rats. Chimeras are generated. Glomerular podocytes and proximal convoluted tubules develop from the mouse ESCs and ureter-bladder junctions are formed in the chimeric rats.

1. Blastocyst complementation is not new although it has to my knowledge not been successfully used to generate mouse kidney tissue in the rat. Belmonte's group has recently published (2016) generation of human organs in pigs using this approach.
2. Page 2. The authors should be more clear about why the *Sall1*^{+/mut} fetuses exhibited tdTomato fluorescence.
3. Of 18 *Sall1*^{mut/mut} chimeric rat fetuses 12 had a pair of kidneys complemented with mouse ESCs. In another experiment the authors evaluated if the chimeric animals survived. None of the animals survived. This mortality was attributed to the role of *Sall1* in suckling but there is no confirmation that this is the case. Were there postmortem studies on the mice to evaluate whether there was no ureteral functional or anatomical obstruction?
4. Figure 1b. The *Sall1*^{+/mut} mice do not look normal although they look better than the *Sall1*^{mut/mut} mice. The authors should have shown a *Sall1*^{+/+} kidney also.

5. Figure 2d. Is there any incorporation of GFP into *Sal1*^{+/+} fetuses?
6. Figure 2e. Is there patency proven between the ureter and the bladder of the chimeras?
7. Figure 2f. Again the authors indicate normality of the ESC-complemented kidneys without showing a control at the same age.
8. Figure 2g. The staining for calbindin is not very clearly specific. Also this protein is expressed in the distal tubule and connecting piece (early part of collecting system). There is no staining for a proximal tubule marker. Again there should be control *Sal1*^{+/+} mice kidneys stained for comparisons.
9. In the chimeras where does the vasculature derive from: mouse or rat?

Reviewer #3:

Remarks to the Author:

The paper by Goto and colleagues demonstrates cross-species complementation of mouse ES cells in rat fetuses – important for xenotransplantation. The study is built on prior studies of blastocyst complementation – whereby genetically sufficient ES cells can contribute to normal embryonic development in genetically deficient animals. The ability of mouse ES cells to complement rat fetal development is exciting. Cross-species complementation has been previously demonstrated for the pancreas (by same investigators), with rat iPSCs contributing to pancreas development.

The paper would be strengthened by the following:

- 1) Discussion of factors that promote successful complementation (what permits complementation – between strains and why do certain tissues appear to be complemented more successfully than others?)
- 2) What is the reason for difference in complementation from mouse to rat vs. rat to mouse? Is this due to time of cell division or some other feature? Even some speculative or proposed mechanisms would be helpful.
- 3) Timing/cell division between species – is this important? Are the kidneys the size of rat kidneys? Additional evaluation timepoints would be valuable.
- 4) Why don't the kidneys support survival greater than 4 days? Only one mut/mut rat survived to 4 days – unclear if reported perinatal death is due to renal or extrarenal effects. If the latter, it would be very interesting to determine if a renal-specific *Sal1* deficient rat can be rescued by blastocyst complementation from mouse ES cells – and if these kidneys support longterm survival. Given ability to do Crispr genome editing, it should be possible to generate conditional alleles. It is unclear why Crispr was used to generate additional conventional KO animals rather than conditional lines.
- 5) Additional histology, quantification would be valuable. The images included are difficult to assess.
- 6) Intraurethral dye injection to assess bladder/ureter connections is important. Additional functional analysis (if at all possible) would be very valuable.

We thank you all for your time and effort in reviewing our manuscript **NCOMMS-17-27992-T** entitled "*Generation of pluripotent stem cell-derived mouse kidneys in Sall1-targeted anephric rats*". We have addressed all the concerns raised and have incorporated valuable suggestions by the reviewers. This has strengthened and improved the manuscript. Here is a point-wise response to the reviewer's comments.

Reviewers' comments:

Reviewer #1 (Remarks to the Author):

This manuscript reports the generation of Sall1 mutant rats with impaired nephrogenesis and the use of blastocyst complementation with GFP labeled mouse embryonic stem (ES) cells to generate fetal 3D kidneys with an intact ureter bladder connection. Nephrogenesis was detected by the presence of podocyte and tubules specific markers.

The analysis of E21.5 Sall1^{-/-} chimeric fetuses showed that the glomerular podocytes and proximal convoluted tubules originated from mouse ES cells, while the other parts of the kidney still contained rat cells. This limits the use of this approach for generating kidney grafts escaping immune rejection.

We have extensively revisited the immunohistological analysis by staining functional markers of each renal competent (**Supplementary Table1**) and added the images in the revised **Fig. 4a,b** and **Supplementary Figure 6** with all the proper controls (mouse and rat each). These data indicates that collecting tubes and blood vessels of mouse PSC-derived kidney are composed of a mixture of mouse and rat cells. These facts may cause immune rejection as you mentioned. We mentioned this in the **Discussion** (lines 241-245).

Although kidney formation was detected after xenogeneic blastocyst complementation in SALL1 deficient fetuses, postnatal development of the chimeric animals was apparently hampered, since none of the surviving chimeras had the Sall1^{-/-} genotype. The authors speculate that other effects (not related to kidney development) of Sall1 deficiency, such as impaired suckling may be responsible for postnatal lethality of blastocyst complemented Sall1^{-/-} chimeras. A more thorough analysis of postnatal Sall1^{-/-} chimeras would be required rather than just mentioning remnants of one chimeric offspring retrieved at postnatal day 4.

Upon further analysis of the postnatal *Sall1^{mut/mut}* chimeras, we found that none of the chimeras (n=11) had milk in their stomachs and died of dehydration (**Supplementary**

Figure 5a). In newborn rodents, an inability to suckle is linked to anosmia¹, importantly, it was reported that the olfactory bulb is severely impaired in *Sall1* mutant mice² likely rendering them anosmic³. We have added this conjecture along with the relevant references and provided a potential solution to overcome this problem in the current **Discussion** (lines 232-240), along with **Results** (lines 162-165).

A previous study (Usui et al., Am J Pathol 180, 2417–2426, 2012) reported that it was not possible to generate in SALL1 deficient mice kidneys from pluripotent rat stem cells. The authors speculated that key molecules involved in the interactions of the metanephric mesenchyme and ureteric buds do not crossreact between mouse and rat. The present study raises the question why it works – at least to some extent – the other way round. Potential mechanisms should be discussed.

To clarify why rat PSCs cannot complement *Sall1*^{mut/mut} mouse in our previous study⁴, we investigated the contribution of PSCs to the developing kidney in interspecific chimeras between mouse and rat. We found that rat PSCs contribute to the metanephric mesenchyme, a nephron progenitor, less efficiently than mouse PSCs (**Fig. 1**). In addition, we found that low chimerism of mouse PSCs is highly correlated with failure of the kidney complementation in *Sall1*^{mut/mut} rats (**Fig. 3c**). Thus, we conclude that a minimum level of PSC contribution to the metanephric mesenchyme is key for successful complementation of the kidney. Given the new data, we have revised the **Result** (lines 76-97) and **Discussion** (lines 207-225).

Since the introduction of the manuscript starts with the problem of organ transplantation, the discussion should include potential pathways how the approach might be further developed towards clinical application. This would require host embryos from larger species, such as pig. Previous studies involving complementation of porcine blastocysts with human pluripotent stem cells showed very low chimerism. Are there strategies how to overcome this problem?

We would like to thank the Reviewer for this suggestion. We now discuss the potential considerations of the Blastocyst complementation approach for clinical application in the **Discussion** (lines 248-259). We mainly delve on 3 points; (1) Requirement of eliminating host-derived cells, in particular the collecting tube and blood vessels which are a mixture of donor- and host-derived cells, in this study (2) Necessity to increase chimerism of human cells in livestock animal, (3) Also limiting the potency of PSCs, to differentiate into germline

and neural cells, which raises ethical concerns. We hope this paragraph provides readers a broader perspective on the future of this approach.

Reviewer #2 (Remarks to the Author):

The authors describe that mouse embryonic stem cells injected into rat blastocysts can rescue kidney development when the blastocysts are derived from Sall1 knockout rats. Chimeras are generated. Glomerular podocytes and proximal convoluted tubules develop from the mouse ESCs and ureterbladder junctions are formed in the chimeric rats.

1. Blastocyst complementation is not new although it has to my knowledge not been successfully used to generate mouse kidney tissue in the rat. Belmonte's group has recently published (2016) generation of human organs in pigs using this approach.

In the study by Belmonte's group⁵, they do not show successful generation of entire organs in the pig by interspecific blastocyst complementation using human PSCs, and only show a low level of contribution (less than 0.1%) of human PSCs to pig tissues.

2. Page 2. The authors should be more clear about why the Sall1+/mut fetuses exhibited tdTomato fluorescence.

To clarify the differences between the two mutant strategies were as follows: one was made by microinjection of CRISPR/Cas9 system to cause a large deletion of *Sall1* gene, and the other was made by homologous recombination in rat ESCs to replace *Sall1* gene with *tdTomato*, which allows visualization of the expression pattern. We have clearly distinguished them as *Sall1*^{mut/mut} and *Sall1*^{tdTomato/tdTomato}, and revised the **Result** (lines 100-127) and **Fig. 2**.

3. Of 18 Sall1mut/mut chimeric rat fetuses 12 had a pair of kidneys complemented with mouse ESCs. In another experiment the authors evaluated if the chimeric animals survived. None of the animals survived. This mortality was attributed to the role of Sall1 in suckling but there is no confirmation that this is the case. Were there postmortem studies on the mice to evaluate whether there was no ureteral functional or anatomical obstruction?

We confirmed none of the chimeras had milk in their stomachs, suggesting their inability of

milk suckling (**Supplementary Figure 5a**). For a postmortem study, to check if the filtering system in the kidneys work, we measured blood urea nitrogen (BUN) and creatinine (CRE) in the serum of *Sall1^{mut/mut}* chimeric rats with mouse PSC-derived kidney, 10-14 hours after birth. However, no significant difference compared with *Sall1^{+/+}* and *Sall1^{+mut}* chimeras as well as *Sall1^{mut/mut}* rats lacking kidneys was observed (**Supplementary Figure 5b**). To assess the renal function clearly, further activity of the neonates postnatally is essential, which cannot be obtained in the present study. Instead, we have added new data showing mouse PSC-derived renal components express important functional markers (**Fig. 4a**). We also were able to show patency between the ureter and the bladder (**Fig. 4d**), which makes a strong case for potential functioning of the kidney. With regard to the inability of *Sall1^{mut/mut}* rats to suckle milk at birth, the reason is discussed in lines 232-240 and pertains to anosmia in the mutants.

4. Figure 1b. The Sall1+/mut mice do not look normal although they look better than the Sall1mut/mut mice. The authors should have shown a Sall1+/+ kidney also.

In the revised **Fig. 2d**, we have added a section of a kidney in *Sall1^{+/+}* rat showing similar histology to *Sall1^{+mut}* rat, suggesting that they are normal.

5. Figure 2d. Is there any incorporation of GFP into Sall1+/+ fetuses?

Yes, there is. In the revised **Fig. 3d**, we have added a representative picture of *Sall1^{+/+}* rat fetus with mouse ESC contributions.

6. Figure 2e. Is there patency proven between the ureter and the bladder of the chimeras?

Yes there is. We show evidence for that in the manuscript (**Fig. 4d**).

7. Figure 2f. Again the authors indicate normality of the ESC complemented kidneys without showing a control at the same age.

We regret not providing a suitable control. We have added it in the revised **Fig. 3e**.

8. Figure 2g. The staining for calbindin is not very clearly specific. Also this protein is expressed in the distal tubule and connecting piece (early part of collecting system). There is no staining for a proximal tubule marker. Again there should be control Sall1+/+ mice

kidneys stained for comparisons.

We have extensively revisited the immunohistological analysis by staining functional markers of each renal competent (**Supplementary Table1**) and added the images in the revised **Fig. 4a** and **Supplementary Figure 6** with all the proper controls (mouse and rat each).

9. In the chimeras where does the vasculature derive from: mouse or rat?

The vasculature consists of both mouse and rat cells. We have added this data in the revised **Fig. 4b**.

Reviewer#3 (Remarks to the Author):

The paper by Goto and colleagues demonstrates crossspecies complementation of mouse ES cells in rat fetuses – important for xenotransplantation. The study is built on prior studies of blastocyst complementation – whereby genetically sufficient ES cells can contribute to normal embryonic development in genetically deficient animals. The ability of mouse ES cells to complement rat fetal development is exciting. Crossspecies complementation has been previously demonstrated for the pancreas (by same investigators), with rat iPSCs contributing to pancreas development.

We thank the reviewer for the encouraging and supportive comments.

The paper would be strengthened by the following:

1) Discussion of factors that promote successful complementation (what permits complementation – between strains and why do certain tissues appear to be complemented more successfully than others?)

2) What is the reason for difference in complementation from mouse to rat vs. rat to mouse? Is this due to time of cell division or some other feature? Even some speculative or proposed mechanisms would be helpful.

To clarify why rat PSCs cannot complement *Sall1^{mut/mut}* mouse in the previous study, we investigated the contribution of PSCs to the developing kidney in interspecific chimeras between mouse and rat. We found that rat PSCs contribute to metanephric mesenchyme, less efficiently than mouse PSCs (**Fig. 1**). In addition, we found low chimerism of mouse

PSCs is highly correlated with failure of the kidney complementation in *Sall1^{mut/mut}* rats (**Fig. 3c**). Thus, we concluded a minimum level of PSC contribution to the metanephric mesenchyme is key for successful complementation of the kidney. Given the new data, we have revised the **Result** (lines 76-97) and **Discussion** (lines 207-225).

3) Timing/cell division between species – is this important? Are the kidneys the size of rat kidneys? Additional evaluation timepoints would be valuable.

Timing/cell division between species would be very important when using more disparate species. Regarding the size of the kidney, interestingly, mouse PSC-derived kidneys generated in rat were smaller than normal neonatal and *Sall1^{+mut}* and *Sall1^{+/+}* chimeric rats, and were similar to that of mice kidneys (**Fig. 3f**). We have revised the manuscript accordingly and added our hypothesis (lines 153-157).

*4) Why don't the kidneys support survival greater than 4 days? Only one mut/mut rat survived to 4 days – unclear if reported perinatal death is due to renal or extrarenal effects. If the latter, it would be very interesting to determine if a renal specific *Sal1* deficient rat can be rescued by blastocyst complementation from mouse ES cells –and if these kidneys support longterm survival. Given ability to do Crispr genome editing, it should be possible to generate conditional alleles. It is unclear why Crispr was used to generate additional conventional KO animals rather than conditional lines.*

We confirmed none of the chimeras had milk in their stomachs, suggesting the inability of suckling caused the lethality (**Supplementary Figure 5a**). As the reviewer suggested, potential use of alternative models such as kidney specific deletion of a targeted gene should be an important consideration future strategies, and we now added it into the **Discussion** (lines 237-240).

5) Additional histology, quantification would be valuable. The images included are difficult to assess.

We have revised the histological data by providing clearer images. In the revised **Fig. 1a** showing the difference of the contribution of mouse and rat PSCs in the interspecific chimera, we quantified the contribution of xenogenic PSCs to *Sall1* positive metanephric mesenchyme by counting 2 sections from multiple embryos (4 in R->M and 3 in M->R respectively) using image J software (**Fig. 1b** and **Methods**).

6) Intraurethral dye injection to assess bladder/ureter connections is important. Additional functional analysis (if at all possible) would be very valuable.

We would like to thank you for highlighting the importance of showing connections between kidney and bladder/ureter. We have duly revised the manuscript and added this data in **Fig. 4d**. For additional functional analysis, we confirmed the expression of functional markers of renal components (**Fig. 4a**). Also, we measured blood urea nitrogen (BUN) and creatinine (CRE) in the serum of *Sall1^{mut/mut}* chimeric rats with mouse PSC-derived kidney, 10-14 hours after birth to check if the filtering system in the kidneys work. However, no significant difference compared with *Sall1^{+/+}* and *Sall1^{+/mut}* chimeras as well as *Sall1^{mut/mut}* rats lacking kidneys was observed (**Supplementary Figure 5b**). To assess renal function better, adequate postnatal activity of the neonates would be desirable, which is unfortunately limited by the *Sall1* mutant phenotype.

REFERENCES

1. Mori K, Nagao H, Yoshihara Y. The olfactory bulb: coding and processing of odor molecule information. *Science* **286**, 711-715 (1999).
2. Harrison SJ, Nishinakamura R, Monaghan AP. *Sall1* regulates mitral cell development and olfactory nerve extension in the developing olfactory bulb. *Cereb Cortex* **18**, 1604-1617 (2008).
3. Elling U. Genetic Analysis of the *Sall* Transcription Factor Family in Murine Development. Thesis, University of Regensburg <https://epub.uni-regensburg.de/10476/1/Elling-P%20H%20D.pdf>, (2005).
4. Usui J, Kobayashi T, Yamaguchi T, Knisely AS, Nishinakamura R, Nakauchi H. Generation of kidney from pluripotent stem cells via blastocyst complementation. *Am J Pathol* **180**, 2417-2426 (2012).
5. Wu, J. et al. Interspecies chimerism with mammalian pluripotent stem cells. *Cell* **168**, 473-486 (2017).

Reviewers' Comments:

Reviewer #1:

Remarks to the Author:

Thank you for addressing my comments.

Reviewer #2:

Remarks to the Author:

The authors have adequately addressed my concerns.

Reviewer #3:

Remarks to the Author:

The authors have made a number of substantial improvements in the revised version of the manuscript. The finding that mouse PSCs can contribute significantly to rescue the kidney phenotype in a rat knockout is exciting.

1. While the explanation of the inefficiency of rat PSCs to rescue the kidney phenotype in a mouse *Sal1* mutant is helpful, it does not explain the underlying mechanism. Why should rat PSCs contribute more poorly to metanephric mesenchyme in a mouse than vice versa? Is it because they divide more slowly (i.e. a species-issue) or is it a defect in the specific line used? i.e. how many rat PSC lines were tested? (arising from independent clones). As the interspecies complementation is central to the paper, it is important to provide more information regarding this key component.
2. The contribution of mouse PSCs to many lineages of the kidney in rat recipient is valuable information; some higher resolution demonstration of correct localization of markers (e.g. distribution of nephrin, transporters to correct apical or basolateral location) would be helpful.
3. Additional images (electron micrographs) and higher resolution histology would help to determine how intact the kidneys are in the absence of functional data.
4. The smaller size of kidney is intriguing – given the normal shape (although it looks as if there is sometimes unilateral variation in size – which would be helpful to quantify). Are the glomeruli also smaller in size? Podocyte counting would be of interest – this suggests intrinsic size regulation.
5. Blood urea and creatinine is difficult to interpret (and measure) – is there any urine in the bladder at birth? Analysis of the urine would be of more value if possible.
6. A very powerful experiment – as the authors agree – remains a rescue of a kidney-specific knockout of *Sal1* to prevent early lethality from extrarenal effects. It is still not clear from current paper whether the kidney is functional or would be functional if the mice survived without suckling defects.

We thank you all for your time and effort in reviewing our manuscript **NCOMMS-17-27992-A** entitled "*Generation of pluripotent stem cell-derived mouse kidneys in Sall1-targeted anephric rats*". We are glad to hear full satisfaction of reviewers #1 & #2 for our revised manuscript. We have addressed to all the concerns and valuable suggestions of the reviewer #3. We believe these changes (highlighted in the revised manuscript) have strengthened and improved our manuscript. You can see our point-by-point responses below.

Reviewers' comments:

Reviewer #3 (Remarks to the Author):

The authors have made a number of substantial improvements in the revised version of the manuscript. The finding that mouse PSCs can contribute significantly to rescue the kidney phenotype in a rat knockout is exciting.

1. While the explanation of the inefficiency of rat PSCs to rescue the kidney phenotype in a mouse Sal1 mutant is helpful, it does not explain the underlying mechanism. Why should rat PSCs contribute more poorly to metanephric mesenchyme in a mouse than vice versa? Is it because they divide more slowly (i.e. a species-issue) or is it a defect in the specific line used? i.e. how many rat PSC lines were tested? (arising from independent clones). As the interspecies complementation is central to the paper, it is important to provide more information regarding this key component.

Recently our group reported chimerism in interspecific chimera between mouse and rat varies organ-to-organ than in allogeneic chimera (Yamaguchi *et al.*, **Sci Rep** (2018) doi: 10.1038/s41598-018-34193-1). Importantly, while some organs including kidney showed lower chimerism in R->M but higher in M->R, the other organs such as heart, lung and intestine showed opposite trends (please see a figure attached below from the paper). We tested 2 independent rat ESC lines and the two lines showed consistent results. Thus, we think the variability of the chimerism is likely due to environmental cues depending on organs and tissues rather than proliferation and potency of the PSCs. We discuss above with a citation (Yamaguchi *et al.*, **Sci Rep** (2018)) in Line 220-230).

Figure 3. Contributions of rat or mouse PSCs to the organs of interspecies chimeras Donor PSC derivative chimerism in brain, CD45⁻ hepatic cells in fetal liver (FL CD45⁻), heart, lung, intestine, embryonic fibroblast (EF), CD45⁺ hematopoietic cells in fetal liver (FL CD45⁺), kidney, and SSEA1⁺ gonad cells (Gonad SSEA1⁺) of E14 to E15 chimeras. (A) Chimerism of rat ESC derivatives in the organs of intraspecies chimeras (left; n = 6), chimerism of rat iPSC derivatives in the organs of interspecies chimeras (middle; n = 10), and chimerism of rat ESC derivatives in the organs of interspecies chimeras (right; n = 10). (Mean values ± SEM were obtained from 6 or 10 independent experiments. *P < 0.05; Student's *t*-test.) (B) Chimerism of mouse ESC derivatives in the organs of intraspecies chimeras (left; n = 5), chimerism of mouse iPSCs derivatives in the organs of interspecies chimeras (middle; n = 5), and chimerism of mouse ESCs derivatives in the organs of interspecies chimeras (right; n = 23 to 28). (Mean values ± SEM were obtained from 5, 23, or 28 independent experiments. *P < 0.05; Student's *t*-test.)

2. The contribution of mouse PSCs to many lineages of the kidney in rat recipient is valuable information; some higher resolution demonstration of correct localization of markers (e.g. distribution of nephrin, transporters to correct apical or basolateral location) would be helpful.

According to the reviewer's suggestion, we revised figure 4 as to be a higher resolution. Dotted lines in figure 4 and Supplementary figure 6 were added to make the renal components clear. Line 179-181 in the result section and Line 584-587 in the figure legend

explain this revision.

3. Additional images (electron micrographs) and higher resolution histology would help to determine how intact the kidneys are in the absence of functional data.

Unfortunately, we don't have any fixed samples for electron micrographs at this time point. Therefore, additional TEM analysis is too time-consuming to complete. Instead, we believe higher resolution histology in revised Figure 4 helps to clarify the normality of the generated kidney.

4. The smaller size of kidney is intriguing – given the normal shape (although it looks as if there is sometimes unilateral variation in size – which would be helpful to quantify). Are the glomeruli also smaller in size? Podocyte counting would be of interest – this suggests intrinsic size regulation.

According to reviewer's suggestion, we have added new data regarding the number of glomeruli and the size of individual glomeruli among kidneys in *Sal1*^{mut/mut} chimeric rats, control mice and rats (Line 191-194 and Supplementary Fig. 7).

5. Blood urea and creatinine is difficult to interpret (and measure) – is there any urine in the bladder at birth? Analysis of the urine would be of more value if possible.

Due to the very limited volume of urine recovered, we actually failed to measure the urinary levels of BUN and CRE. Blood levels of BUN and CRE had been posted instead. Successful maturation of the pups with generated kidneys using alternative model as discussed in Line 242-245 would allow us to fully address the function in the future study.

*6. A very powerful experiment – as the authors agree – remains a rescue of a kidney-specific knockout of *Sal1* to prevent early lethality from extrarenal effects. It is still not clear from current paper whether the kidney is functional or would be functional if the mice survived without suckling defects.*

We strongly agree with the reviewer's opinion that it is really important to rescue a kidney-specific knockout of *Sal1* or alternative gene(s) to prevent the early lethality, as we discussed in Line 242-245. Seeking more appropriate model for kidney generation is beyond the scope of this study, but we would like to overcome the current limitation in the

future study.

Reviewers' Comments:

Reviewer #3:

Remarks to the Author:

The authors have provided additional data to support their findings. It will be very interesting to know in future, if a conditional deletion - with renal-restricted defects - can be rescued with a functional kidney. It will also be interesting to know specifics about species-specific regulation of organ size and nephron endowment - this will also require additional quantification and experiments in future studies.